# Distal Humeral Replacement in Patients with Primary Bone Sarcoma: The Functional Outcome and Return to Sports

**DOI:** 10.3390/cancers15133534

**Published:** 2023-07-07

**Authors:** Kristian Nikolaus Schneider, Moritz Ellerbrock, Georg Gosheger, Lucia Maria Westphal, Niklas Deventer, Sebastian Klingebiel, Carolin Rickert, Christoph Theil

**Affiliations:** Department of Orthopaedics and Tumor Orthopaedics, Albert-Schweitzer Campus 1, 48149 Münster, Germany; moritz.ellerbrock@ukmuenster.de (M.E.); georg.gosheger@ukmuenster.de (G.G.); lwestpha@uni-muenster.de (L.M.W.); niklas.deventer@ukmuenster.de (N.D.); sebastian.klingebiel@ukmuenster.de (S.K.); carolin.rickert@ukmuenster.de (C.R.); christoph.theil@ukmuenster.de (C.T.)

**Keywords:** megaprosthesis, elbow, humerus, distal, bone sarcoma, return to sport, functional outcome

## Abstract

**Simple Summary:**

Primary bone sarcomas around the distal humerus are rare. Following tumor resection, distal humeral replacement (DHR) is a limb-salvage option for the reconstruction of the resulting bone defect. Due to small patient numbers, the postoperative functional outcome and resumption of sporting activities among affected patients remain mostly unclear although they become increasingly important with improvements in patients’ life expectancy and quality of life. Thus, the aims of our study were to evaluate the functional outcome, assess the return to sport activities, and identify the potential limiting and beneficial factors in a large homogenous patient cohort treated with a single-design modular implant at a tertiary sarcoma center.

**Abstract:**

Distal humeral replacement (DHR) is a limb-salvage option for the endoprosthetic reconstruction of bone defects following the resection of a primary bone sarcoma. As primary bone sarcomas are only occasionally located around the distal humerus, there is a paucity of information regarding postoperative function, and patients’ resumption of sporting activities. With advances in diagnostics and in surgical and oncological treatment leading to an increased patient life expectancy and higher quality of life, patients’ functional outcome and return to sports activities are of increasing interest. Between 1997 and 2021, a total of 24 patients underwent DHR with a single-design modular implant at a tertiary sarcoma center. A total of 14 patients who died of their disease were excluded, leaving a study cohort of 10 patients, with a median age of 30 years on the day of surgery (IQR 20–37). At the last follow-up, after a median of 230 months (IQR 165–262), the median MSTS was 19 (IQR 13–24), the median TESS was 79 (IQR 66–87), the median SEV was 38% (IQR 24–53), the median TS was 6 (IQR 4–7), and the median WAS was 3 (IQR 1–8). Among the variables of gender, surgery on the dominant extremity, intraoperative nerve resection, extra-articular tumor resection, chemotherapy, radiotherapy, and revision surgeries, none were associated with a better/lower functional outcome score or return to sports activities. However, a higher level of sports performance prior to diagnosis (WAS > 10) was associated with a higher level of sports performance postoperatively (*p* = 0.044).

## 1. Introduction

Most sarcomas require resection with tumor-free margins as a part of multimodal therapy [1,2]. For sarcomas involving the metaphysis, this usually results in a juxta-articular bone defect that can be reconstructed using modular megaprosthesis [3,4,5]. While these implants provide immediate stability and allow for early rehabilitation, implant failures and associated revision surgeries are very common in elbow reconstructions [3,6,7]. Due to the rarity of distal humerus megaprosthetic reconstruction, and the high likelihood of complications such as implant failure or elbow stiffness, the data on the postoperative functional outcome and the potential return to sports activities are limited [6,7,8,9]. Continuous improvements in diagnostics, surgical, and oncological treatment have led to an increased patient life expectancy [10,11]. Thus, for long-term surviving patients, the functional outcome and participation in sports are growing in importance [10,12,13,14]. While good-to-excellent functional outcomes are reported for megaprosthetic reconstructions around the proximal humerus, the data regarding reconstructions around the distal humerus are scarce, and the limited information available is often derived from small and heterogenous patient cohorts, involving primary and secondary malignancies, and/or different types of reconstruction and implant system [4,6,7,10,15,16]. Thus, the aims of our study were to (1) evaluate the functional outcome, (2) assess the return to sports activities, and (3) identify potential limiting and beneficial factors in a large homogenous patient cohort treated with a single-design modular implant at a tertiary sarcoma center. 

## 2. Materials and Methods

Between September 1997 and November 2021, a total of 958 patients underwent the resection of a primary bone sarcoma or locally aggressive and rarely metastasizing bone tumor, and reconstruction with a single-design modular implant, at a tertiary university hospital (Figure 1). In 24 (2.5%) patients, the location was the distal humerus. A total of 14 patients died of their disease and were excluded, leaving 10 patients as the final study cohort (Table 1). The study was approved by the local ethics committee (reference number 2018-199-f-S), and performed in accordance with the Declaration of Helsinki. 

### 2.1. Surgical and Implant Details

Megaprosthetic reconstruction, as part of multimodal therapy, was performed as the treatment of choice for distal humerus sarcomas and recurrent giant cell tumors with joint involvement, after failed intralesional resection. We did not perform biological reconstructions for this indication. For all surgeries, we used the MUTARS (Modular Tumor and Revision System, Implantcast, Buxtehude, Germany) implant, which uses a fixed hinge elbow joint (Figure 2). 

### 2.2. Data Collection

Data regarding demographics, as well as surgical and oncological details, were obtained from the patients’ electronic medical records. 

### 2.3. Functional Outcome

The functional outcome was assessed using two standardized scoring systems, specifically designed to assess the functional outcome following limb-sparing surgery in musculoskeletal oncology: the Musculoskeletal Tumor Society Score (MSTS), and the Toronto Extremity Salvage Score (TESS). The upper-extremity version of the MSTS is a six-item questionnaire covering dexterity, pain, emotional acceptance, function, hand positioning, and lifting abilities. Each question is scored on a scale from 0 (very limited) to 5 (no restriction), with a maximum score of 30 [17]. The upper-extremity version of the TESS is a 29-item questionnaire covering everyday tasks involving the upper extremity, with each scored on a scale from 1 (impossible to do) to 5 (not at all difficult), with a maximum score of 145 converted into a percentage as the final score [18]. In addition, the subjective elbow value (SEV), a patient-reported percentage of the elbow function compared to the non-affected elbow on a scale from 0% to 100% (no side-to-side difference) was obtained [19]. 

### 2.4. Return to Sports Activities

Participation in sports activities was also evaluated, using standardized scoring systems: the Tegner Activity Scale (TS), and the modified weighted activity score (WAS). The TS assesses the performed level of sports activities on a scale from 1 (cannot move) to 10 (competitive contact sports on a national elite level) [20]. The WAS summarizes patients’ individual performance, by multiplying the frequency (sessions per week), duration (in hours), and impact of each performed sports activity (low-impact sports activity = 1, high-impact sports activity = 3; Table 2) [21,22]. The final WAS is obtained by adding the individual scores of each sports activity, whereby a WAS of 0–10 marks a low-activity patient, and a WAS of >10 denotes a high-activity patient [10,21]. 

### 2.5. Statistical Analysis

Statistical analyses were performed using SPSS 25.0 (IBM Corp., Armonk, NY, USA). The data distribution was determined using the Kolmogorov–Smirnov test. Parametric analyses were performed using Student’s *t*-test, and non-parametric analyses were performed using the Mann–Whitney U test. All *p*-values were two-sided, and a *p*-value of <0.05 was considered statistically significant. 

## 3. Results

At the last follow-up, after a median of 230 months (IQR 165–262), the results from ten patients (100% follow-up rate for survivors, five female), with a median age of 30 years at surgery (IQR 20–37), and a median BMI of 23 (IQR 20–24), were available for analysis.

### 3.1. Functional Outcome

Prior to diagnosis, the median MSTS was 30 (IQR 23–30), and the median TESS was 100% (IQR 66–87). At the last follow-up, the median MSTS was 19 (IQR 13–24), the median TESS was 79% (IQR 66–87), and the median SEV was 38% (IQR 24–53). Postoperatively, the median total range of motion in the elbow joint was 113° (IQR 94–123), with a median extension lag of 20° (IQR 10–31).

### 3.2. Return to Sports Activities

Prior to diagnosis, the median TS was 6 (IQR 5–9), and the median WAS was 5 (IQR 1–8). At the last follow-up, the median TS was 6 (IQR 4–7), the median TS difference (TS prior to diagnosis-TS at last follow-up) was 2 (IQR 1–3), and the median WAS was 3 (IQR 1–8). At the last follow-up, 7 of 10 (70%) patients participated in at least one low-impact type of sports activity (Table 3).

### 3.3. Beneficial and Limiting Factors

Among the variables of gender, surgery on the dominant extremity, intraoperative nerve resection (the ulnar nerve had to be resected in two patients, in order to achieve wide surgical margins) or postoperative nerve palsy (of the radial nerve in two patients), extra-articular tumor resection, chemotherapy, radiotherapy, and a revision surgery, none were associated with a higher/lower functional outcome score, nor a better/lower return to sports activities. However, patients who performed a higher level of sports prior to diagnosis had a better WAS at the final follow-up (Table 4).

## 4. Discussion

The main findings of our study are as follows: (1) the functional outcome following DHR is acceptable; (2) patients regularly return to sporting activities, although the level of performed sports is lower overall than prior to diagnosis; and (3) a number of variables were not associated with better/worse functional outcome scores and/or higher/lower rates of return to sports activities. These variables included gender, surgery on the dominant extremity, intraoperative resection of the ulnar nerve or postoperative palsy of the radial nerve, an extra-articular tumor resection, chemotherapy, radiotherapy, or a revision surgery. However, patients who performed a higher level of sports prior to diagnosis were more likely to perform higher level of sports postoperatively.

The functional outcome scores following DHR were acceptable, but lower than those of previous studies: Funovics et al. reported a mean MSTS of 24, and a mean follow-up of 28 months in 27 patients undergoing DHR with a single-design modular implant [7]. However, only 38% of their patients were treated for primary sarcomas; 62% of their patients were treated for metastatic bone disease, and underwent a marginal tumor excision “without compromising neurovascular structures”, rather than a wide tumor resection, making it difficult to compare these functional results to our cohort [7]. Nearly perfect functional outcome scores with an MSTS of 28 after a mean follow-up of 26 months were reported by Liang et al. for a DHR with a 3D-printed prosthesis, following resection of aggressive or malignant tumors involving the distal humerus (*n* = 8) or proximal ulna (*n* = 5) in 13 patients with a mean age of 48 years [15]. However, their nearly excellent functional outcome scores might be attributable to the fact that a hemiarthroplasty, rather than a total arthroplasty, was used [15]. The findings in a study by Casadei et al. were similar to the results of the present study; they reported a mean MSTS of 22 in 47 patients, after a mean follow-up of 52 months [8]. However, their patient cohort was heterogenous, including primary and secondary malignancies, as well as seven total humerus replacements and twenty-two standard elbow prostheses. In addition, two different modular implant systems, as well as two different standard elbow prostheses, were used and, in sixteen patients, the standard prosthesis was also combined with an allograft [8]. While the small patient cohorts, non-oncological indications, different types of reconstructions, and varying implant systems might be the result of the rarity of elbow tumors, the heterogeneity in their management reflects the lack of an optimal reconstructive technique in this complex situation. This dilutes the generalizability of these findings, and limits the overall conclusions that can be drawn from DHR in oncological patient cohorts [8].

Only a few studies have investigated the resumption of sporting activities following megaprosthetic reconstructions in oncological patient cohorts, although they have become increasingly important with the continuous improvements in patient life expectancy and quality of life [10,12,13]. For the upper extremity, Lang et al. reported a return-to-sport rate of 100% in 18 patients who had undergone a proximal humeral replacement (PHR) after the resection of a primary bone sarcoma [13]. For the same indication, Ellerbrock et al. reported a return-to-sport rate of 69%, a median TS of 5 (IQR 4–6), and a median WAS of 5 (IQR 0–10), in 32 patients [10]. The overall lower functional outcome scores, and a lower return-to-sport rate after DHR, reflect the complexity of megaprosthetic reconstructions around the distal humerus, an observation that is equivalent to the experience of non-oncological patient cohorts of undergoing a hemi- or total arthroplasty of the elbow joint [23,24,25]. In addition, when assessing the return to sports following DHR, we must acknowledge that the majority of postoperative performed sports activities, such as cycling, hiking, and running (Table 3) do not necessitate specific elbow functions, but can be performed with hardly any elbow function at all.

Regarding the beneficial and limiting factors, we were able to show that a higher level of sports performance prior to diagnosis was associated with a higher level of sports performance postoperatively. In line with these findings are the results of Ellerbrock et al., who were also able to show this association for patients undergoing PHR after the resection of a primary bone sarcoma [10]. Interestingly, neither the resection of the ulnar nerve (*n* = 2), nor a postoperative radial nerve palsy (*n* = 2) was associated with lower functional outcome scores, or with a lesser return to sports activities in our cohort. This indicates possible compensational mechanisms, and patients selecting alternative sports activities to overcome postoperative nerval restrictions. Unfortunately, no previous study has yet investigated the effect of nerve resections/postoperative palsies on the functional outcome and return to sports activities in patients undergoing DHR following a wide tumor resection, meaning that our findings and conclusion are hypothetical, and could be proven wrong with higher patient numbers.

We acknowledge several limitations to our study: (1) We performed a retrospective analysis, and were able to include only a small number of patients. Therefore, individual results may greatly impact the reported results, and our findings are prone to bias. However, our study represents the largest and most homogeneous patient cohort treated with a single-design implant at a single institution, reflecting the rarity of these sarcomas and this procedure. (2) As with previous studies on the functional outcome of megaprosthetic reconstructions, the patients included in this study are long-term survivors. This means that the functional outcome, in terms of the quantity and quality of sports activities, might be biased, and the present study’s findings might not be applicable to patients with a shorter rehabilitation, or those still undergoing adjuvant therapy [10]. Furthermore, 14 patients had to be excluded from our study cohort, as they have already died and, unfortunately, no information regarding their individual return to sports was available. This means that the results presented here are likely to be high-end estimates that need to be interpretated with caution. Considering that the majority of patients returned to some degree of sports, future studies should prospectively investigate how this aspect of the postoperative outcome develops over time, and ideally investigate how to enable patients to return to a safe degree of sports, as part of a healthy lifestyle. (3) While we used common scores to evaluate the return to sports and overall activity, these scores have not been validated for this specific patient cohort with megaprostheses following tumor resection. However, as they measure a subjective degree of sports activity, we believed that they were well-suited to answer our study’s questions. Nonetheless, for some questions, it might be interesting to use more objective measuring tools, such as accelerometers [26].

## 5. Conclusions

DHR following the resection of a primary bone sarcoma results in acceptable functional outcome scores, and a regular return to sports activities. However, the level of performed sports is overall lower than prior to diagnosis, and the majority of postoperatively performed sports activities do not require actual elbow function, but can be performed with only limited elbow movements. Among the variables of gender, surgery on the dominant extremity, intraoperative nerve resection or postoperative nerve palsy, extra-articular tumor resection, chemotherapy, radiotherapy, and revision surgeries, none were associated with better/lower functional outcome scores, or a better/worse return to sports activities. However, a higher level of performance of sports before surgery was associated with a higher level of performance of sports postoperatively.

## Figures and Tables

**Figure 1 cancers-15-03534-f001:**
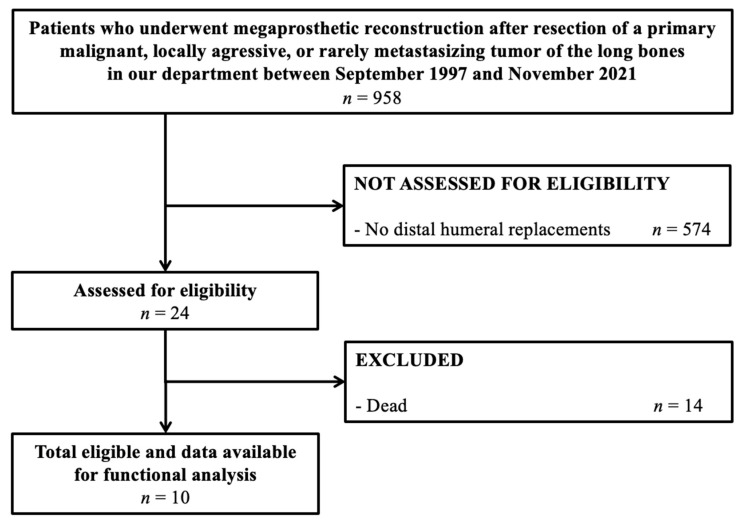
STROBE study flow diagram.

**Figure 2 cancers-15-03534-f002:**
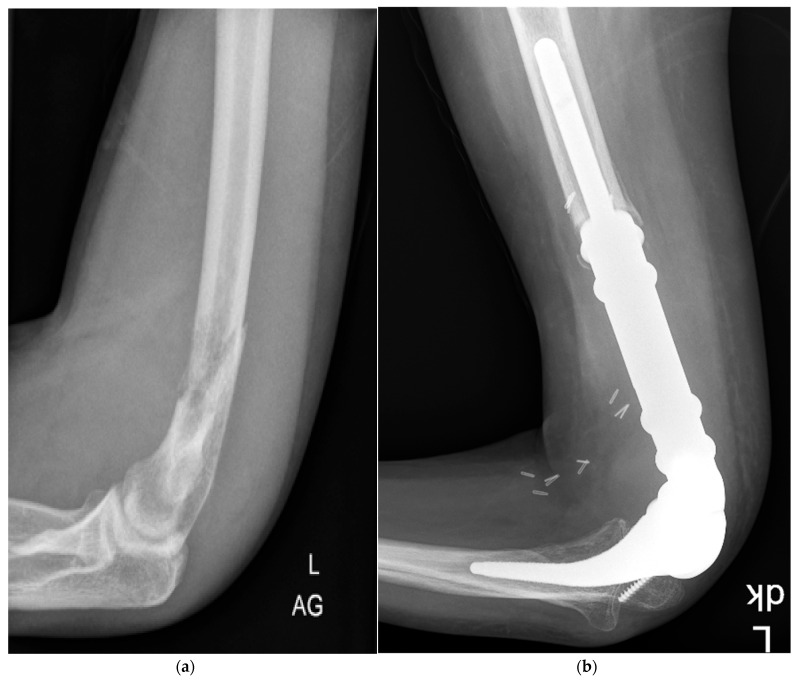
(**a**,**b**) 20-year-old male patient with an osteosarcoma and a pathological fracture of the left distal humerus prior to surgery (**a**), and following wide tumor resection and reconstruction with a modular megaprosthesis (**b**).

**Table 1 cancers-15-03534-t001:** Demographic, surgical, and oncological details.

Variable	*n* (%)
Gender	
Female	4 (40%)
Male	6 (60%)
Tumor entity	
Osteosarcoma	4 (40%)
Ewing sarcoma	3 (30%)
Recurrent giant cell tumor with joint involvement	2 (20%)
NOS	1 (10%)
Affected upper extremity	
Dominant	8 (80%)
Non-dominant	2 (20%)
Intraoperative nerve resection	
Yes	2 (20%)
No	8 (80%)
Type of tumor resection	
Intra-articular	6 (60%)
Extra-articular	4 (40%)
Chemotherapy	
Yes	8 (80%)
No	2 (20%)
Radiotherapy	
Yes	3 (30%)
No	7 (70%)
Revision surgery	
Yes	6 (60%)
No	4 (40%)

**Table 2 cancers-15-03534-t002:** Impact of sports activities according to Healy et al. and Ellerbrock et al. [21,22].

Impact	Type of Sports Activity
Low	Cycling, Nordic walking, hiking
Medium	Running
High	Endurance training, soccer, weightlifting, volleyball, handball

**Table 3 cancers-15-03534-t003:** Sports activities performed by each patient prior to diagnosis, and at the last follow-up.

Patient	Prior Diagnosis	At the Last Follow-Up
1	Cycling	Cycling
2	Swimming, cycling	Cycling
3	Cycling, running, swimming, gymnastics	Cycling, Nordic walking
4	Running, cycling, hiking	Running, cycling, hiking
5	Endurance training, weight lifting	-
6	Volleyball, cycling	Running, cycling
7	Running	-
8	Dancing	Running, hiking
9	Cycling	Cycling
10	Handball, football	-

**Table 4 cancers-15-03534-t004:** Association of demographic and clinical factors with postoperative outcome scores; MSTS = Musculoskeletal Tumor Society Score, TESS = Toronto Extremity Salvage Score; SEV = subjective elbow value; TS = Tegner Activity Score; WAS = weighted activity score; * defined as WAS prior diagnosis > 10; statistically significant *p*-values are in bold.

Factors	Yes	No	*p*-Value
Male	MSTS: 19 (IQR 11–25)	MSTS: 19 (IQR 13–25)	*p* = 1.000
	TESS: 76% (IQR 66–82)	TESS: 83% (IQR 63–88)	*p* = 0.610
	SEV: 30% (IQR 18–53)	SEV: 45% (IQR 33–58)	*p* = 0.352
	TS: 6 (IQR 5–6)	TS: 5 (IQR 4–7)	*p* = 0.914
	WAS: 5 (IQR 1–15)	WAS: 3 (IQR 1–4)	*p* = 0.476
Dominant extremity	MSTS: 19 (IQR 13–25)	MSTS: 19 (IQR 13–24)	*p* = 1.000
	TESS: 80% (IQR 72–87)	TESS: 65% (IQR 58–72)	*p* = 0.267
	SEV: 38% (IQR 26–58)	SEV: 35% (IQR 20–50)	*p* = 0.711
	TS: 6 (IQR 4–7)	TS: 5 (IQR 4–7)	*p* = 1.000
	WAS: 3 (IQR 1–10)	WAS: 4 (IQR 1–7)	*p* = 1.000
Intraoperative nerve resection/postoperative nerve palsy	MSTS: 19 (IQR 13–25)	MSTS: 19 (IQR 11–24)	*p* = 0.762
	TESS: 74% (IQR 60–80)	TESS: 83% (IQR 69–87)	*p* = 0.257
	SEV: 33% (IQR 27–54)	SEV: 45% (IQR 18–53)	*p* = 0.914
	TS: 5 (IQR 4–6)	TS: 6 (IQR 6–7)	*p* = 0.352
	WAS: 8 (IQR 2–21)	WAS: 2 (IQR 0–5)	*p* = 0.171
Extra-articular tumor resection	MSTS: 19 (IQR 13–25)	MSTS: 19 (IQR 11–24)	*p* = 0.762
	TESS: 75% (IQR 61–85)	TESS: 79% (IQR 68–87)	*p* = 0.762
	SEV: 43% (IQR 28–58)	SEV: 35% (IQR 18–53)	*p* = 0.610
	TS: 5 (IQR 4–5)	TS: 6 (IQR 5–6)	*p* = 1.000
	WAS: 8 (IQR 2–21)	WAS: 3 (IQR 0–5)	*p* = 0.257
Radiotherapy	MSTS: 14 (IQR 6–19)	MSTS: 23 (IQR 13–25)	*p* = 0.517
	TESS: 79% (IQR 72–83)	TESS: 79% (IQR 58–87)	*p* = 0.833
	SEV: 20% (IQR 10–35)	SEV: 40% (IQR 30–60)	*p* = 0.183
	TS: 5 (IQR 4–5)	TS: 4 (IQR 4–5)	*p* = 0.667
	WAS: 0 (IQR 0–4)	WAS: 3 (IQR 2–12)	*p* = 0.267
Revision surgery	MSTS: 18 (IQR 11–24)	MSTS: 19 (IQR 14–25)	*p* = 0.476
	TESS: 76% (IQR 58–87)	TESS: 80% (IQR 72–85)	*p* = 0.762
	SEV: 45% (IQR 18–60)	SEV: 33% (IQR 26–46)	*p* = 0.610
	TS: 4 (IQR 4–5)	TS: 6 (IQR 5–6)	*p* = 0.762
	WAS: 3 (IQR 1–5)	WAS: 7 (IQR 1–21)	*p* = 0.610
High level of sports prior to diagnosis *	MSTS: 20 (IQR 14–26)	MSTS: 19 (IQR 13–24)	*p* = 0.400
	TESS: 75% (IQR 69–80)	TESS: 79% (IQR 62–87)	*p* = 0.889
	SEV: 30% (IQR 25–35)	SEV: 45% (IQR 23–58)	*p* = 0.533
	TS: 8 (IQR 7–8)	TS: 5 (IQR 5–6)	*p* = 0.178
	WAS: 18 (IQR 12–24)	WAS: 3 (IQR 0–4)	** *p* ** ** = 0.044**

## Data Availability

All relevant data are contained within this article.

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
