# Peer review of "Distal Humeral Replacement in Patients with Primary Bone Sarcoma: The Functional Outcome and Return to Sports"

_cancers, 2023, doi:10.3390/cancers15133534_

Round 1

Reviewer 1 Report

The manuscript was prepared very well. The introduction section justifies the purpose of the study. I congratulate the authors for the preparation of the manuscript

I would like to congratulate the authors for the structure of the manuscript and all the research carried out. It is highly publishable. However, there are some concerns, in part

Introduction

·       Why is this study considered relevant?

·       I suggest that incorporate more background information on the objective of the study

Methods

·       It would be possible to adjust the study to the CONSORT guidelines to make the publication of your clinical trial completer and more comprehensive. These recommendations are translated into a checklist or verification (checklist) and a flow diagram (flow diagram).

Results

·       It is the strong part of the manuscript does not require any changes

·       Incorporate the abbreviations used in the footers of the tables

Discussion

·       Include a section on strengths / limitations.

·       What does this article contribute to, the authors should make their own assessment and include their own discussion of the results shown in the manuscript?

·       include a section on future scenarios

Conclusion

·        In the Conclusion section, state the most important outcome of your work. Do not simply summarize the points already made in the body — instead, interpret your findings at a higher level of abstraction. Show whether, or to what extent, you have succeeded in addressing the need stated in the Introduction (or objectives).

Author Response

“Introduction

  • Why is this study considered relevant?
  • I suggest that incorporate more background information on the objective of the study”

Answer: We totally agree and have added the respective information to the introduction (line 46, lines 47-49)

“Methods

  • It would be possible to adjust the study to the CONSORT guidelines to make the publication of your clinical trial completer and more comprehensive. These recommendations are translated into a checklist or verification (checklist) and a flow diagram (flow diagram).”

Answer: We totally agree, but the CONSORT guidelines are designed to report on randomized controlled trials. Thus, our study design was adjusted to the STROBE reported which was specifically designed for observational studies. The STROBE diagram is included in our manuscript (Figure 1).

“Discussion

  • Include a section on strengths / limitations.
  • include a section on future scenarios”

Answer: The section “limitations” with discussion of the strength and limitations of our study is already included (lines 243-273). We have added a sentence regarding possible future research (lines 256-268)

“Conclusion

  • In the Conclusion section, state the most important outcome of your work. Do not simply summarize the points already made in the body — instead, interpret your findings at a higher level of abstraction. Show whether, or to what extent, you have succeeded in addressing the need stated in the Introduction (or objectives).”

Answer: We totally agree and have modified the conclusions accordingly (lines 276-279)

Reviewer 2 Report

Congratulations to the authors for the quality of the submitted paper.

Although elbow is a very rare location, the authors collected a case series with interesting numbers (24 patients, with complete follow-up for 10).

The methodology of the study is of excellent quality. The presentation of results and discussion also have an adequate level of depth.

Study worthy of publication, probably for completeness only the introduction could be implemented with more details.

- Line 37-38: emphasize how improved surgical techniques and multimodal treatment (radio, chemo, and immunotherapy) have improved outcomes in the treatment of sarcomas over the past 30 years (PMID: 33261292).

- Line 41: emphasize how in addition to the classic complications expected for this sarcoma surgery, there is also one specific to the "elbow" district and that is stiffness (PMID: 32913596)

Author Response

“Study worthy of publication, probably for completeness only the introduction could be implemented with more details.

- Line 37-38: emphasize how improved surgical techniques and multimodal treatment (radio, chemo, and immunotherapy) have improved outcomes in the treatment of sarcomas over the past 30 years (PMID: 33261292).

- Line 41: emphasize how in addition to the classic complications expected for this sarcoma surgery, there is also one specific to the "elbow" district and that is stiffness (PMID: 32913596)”

Answer: Thank you for these valuable improvements. We have added both references and emphasized the improvements of diagnostics and treatment as well as mentioned the elbow stiffness as a specific complication for these types of reconstructions (lines 47-49)

Reviewer 3 Report

Thank you for allowing me to review this manuscript examining the return to sport for patients undergoing a distal humerus replacement. The group of patients is small (10), how ever that is expected with this surgery. The authors found that a majority of patients were able to return to activities, however they were all low-impact and really don't require the use of the elbow. They found now difference in functional outcomes based on gender or dominant extremity. 

This paper doesn't add too much to the current literature. It shows that patients are still functionally limited with regards to the use of the elbow. The sports they go back too don't require an elbow to "function"

It would be interesting to know if the 14 patients that you excluded who died before your follow-up had returned to any sport.  

Author Response

“This paper doesn't add too much to the current literature. It shows that patients are still functionally limited with regards to the use of the elbow. The sports they go back too don't require an elbow to "function"”

Answer: Thank you for this valuable comment. We totally agree and have specified this within the discussion (lines 224-228)

“It would be interesting to know if the 14 patients that you excluded who died before your follow-up had returned to any sport.”

Answer: This is an important aspect that we have emphasized on within the study’s limitations (lines 252-254).

Round 2

Reviewer 3 Report

Thank you for your changes